# Aragonite dissolution protects calcite at the seafloor

Olivier Sulpis [1✉], Priyanka Agrawal[1], Mariette Wolthers [1], Guy Munhoven[2], Matthew Walker [3,4] & Jack J. Middelburg [1]

In the open ocean, calcium carbonates are mainly found in two mineral forms. Calcite, the least soluble, is widespread at the seafloor, while aragonite, the more soluble, is rarely preserved in marine sediments. Despite its greater solubility, research has shown that aragonite, whose contribution to global pelagic calcification could be at par with that of calcite, is able to reach the deep-ocean. If large quantities of aragonite settle and dissolve at the seafloor, this represents a large source of alkalinity that buffers the deep ocean and favours the preservation of less soluble calcite, acting as a deep-sea, carbonate version of galvanization. Here, we investigate the role of aragonite dissolution on the early diagenesis of calcite-rich sediments using a novel 3D, micrometric-scale reactive-transport model combined with 3D, X-ray tomography structures of natural aragonite and calcite shells. Results highlight the important role of diffusive transport in benthic calcium carbonate dissolution, in agreement with recent work. We show that, locally, aragonite fluxes to the seafloor could be sufficient to suppress calcite dissolution in the top layer of the seabed, possibly causing calcite recrystallization. As aragonite producers are particularly vulnerable to ocean acidification, the proposed galvanizing effect of aragonite could be weakened in the future, and calcite dissolution at the sediment-water interface will have to cover a greater share of $CO_2$ neutralization.

[1] Department of Earth Sciences, Utrecht University, Utrecht, The Netherlands. [2] Département d'Astrophysique, Géophysique et Océanographie, Université de Liège, Liège, Belgium. [3] School of Life Sciences, University of Lincoln, Lincoln, UK. [4] Leeds Institute of Data Analytics (LIDA), University of Leeds, Leeds, UK. ✉email: o.j.t.sulpis@uu.nl

More than a quarter of the Earth's surface is covered by marine sediments rich in calcium carbonate ($CaCO_3$)[1,2], whose dissolution represents the ultimate natural sink for anthropogenic carbon dioxide ($CO_2$)[3]. In the open ocean, that we define here as all oceanic areas beyond continental shelves, most $CaCO_3$ originates from the near surface[4,5], where it is secreted by organisms as building blocks of their shells and skeletons in diverse crystalline structures. Calcite is the most stable $CaCO_3$ mineral under Earth surface conditions[6], and it is believed that calcite accounts for the majority of the oceanic $CaCO_3$ reservoir[7]. There is, however, growing evidence that aragonite, another $CaCO_3$ mineral more soluble than pure calcite[6], could account for a large part of, and even dominate $CaCO_3$ production and cycling[8–11]. In addition, magnesium (Mg) calcites, which can be twice more soluble as aragonite[12], are also thought to be important in the open ocean, secreted by fish[12,13] or imported from shallow shelves and banks[14,15]. While a few global modeling studies have included aragonite[16–18], the majority of existing biogeochemical models used to predict and reconstruct Earth climates treat all $CaCO_3$ as the mineral calcite[19–22].

In the open ocean, aragonite production is dominated by shelled pteropods and heteropods, abundant free swimming sea snails[8,10], and to a lesser extent, by some foraminifera[23] and cold-water coral species[24]. Upon the organisms death, aragonite shells settle through the water column, where they start to dissolve[25] due to (1) internal organic matter degradation[26], (2) their increasing solubility with increasing hydrostatic pressure[27], and (3) the buildup of metabolic $CO_2$ in deep waters[28]. The remaining aragonite deposits at the seafloor. Below the aragonite saturation depth, the depth at which seawater undersaturation with respect to aragonite first occurs and below which aragonite should dissolve, aragonite grains are rarely preserved in sediments[29]. This largely contrasts with calcite, which is commonly found in marine sediments up to several kilometers below the calcite saturation depth[2,30]. That aragonite disappears shallower than calcite in sediments is coherent with aragonite's greater solubility, but why is aragonite not preserved in sediments below its saturation horizon whilst calcite ordinarily is? Potential reasons include the presence of calcite dissolution inhibitors in sediments, or fast aragonite dissolution kinetics, but both are still uncertain or unsupported by recent laboratory experiments[31,32].

Although rarely preserved in sediments, there is clear evidence that aragonite reaches the seafloor even deep below its saturation depth. Sediment traps have recorded high concentrations of pteropod genetic material[33] and suspended aragonite[31] far below the aragonite saturation depth. Thus, a large proportion of settling aragonite grains in the ocean could dissolve at or near the sediment–water interface. Let us now consider a sedimentary system in which calcite and aragonite are both present in seawater undersaturated with respect to both minerals, i.e., a surrogate for a deep-sea sediment. From a thermodynamic perspective, aragonite and calcite should both dissolve, releasing alkalinity and raising $CaCO_3$ saturation states ($\Omega$). Since aragonite is more soluble than calcite, if aragonite dissolution is fast enough, then as long as aragonite is present and dissolving, seawater could remain supersaturated with respect to calcite. As there is nothing to keep seawater saturated with respect to aragonite, since it is the most soluble mineral present, aragonite would eventually fully dissolve. In this conceptual model, the interaction between calcite and aragonite is unidirectional, and the preferential preservation of calcite in sediments is caused by the dissolution of deposited aragonite at the seafloor. This represents a deep-sea, carbonate version of galvanization, in which aragonite sacrifices itself to protect the underlying calcite. In practice, the possible presence of Mg calcites[12,13] at the seafloor could complicate this model

further, and the dissolution of Mg calcites may protect aragonite from dissolution.

Observing aragonite dissolution at the seafloor in situ is difficult because of the limited spatial and temporal resolution of instruments able to reach the deep ocean. Using existing sediment-porewater models is also an imperfect approach, because these models mathematically express grains (e.g., shells) as a spatial continuum of solid[34–36] rather than three-dimensional entities with microstructures and heterogeneities. Thus, existing models are unable to resolve chemical gradients within a single pore, or across the surface of a single grain.

Here, we use a novel three-dimensional model, to simulate dissolution reactions at the micrometer scale for a variety of natural $CaCO_3$ grains virtually placed in seawater (Supplementary Fig. 1), within which chemical reactions, their rates, and transport processes were resolved. The model equations, assumptions, initial conditions and boundary conditions for each simulation are described in the "Methods" section. We demonstrate that molecular diffusion generates large disparities in dissolution rates across mineral surfaces within a single $CaCO_3$ shell, which may account for part of the disagreement among published empirical $CaCO_3$ dissolution rate laws. Then, we simulate the dissolution of an aragonite pteropod shell sitting on top of a calcite sediment bed in a typical deep-sea setting, and show that aragonite dissolution indeed exerts a galvanizing action by favouring the preservation of surrounding calcite particles.

## Results and discussion

**Heterogeneous dissolution of $CaCO_3$ shells**. Most experimental assessments of $CaCO_3$ dissolution rates in seawater to date have measured bulk dissolution rates, by computing dissolution rates from a mass or water-chemistry change over a given amount of time[37–40]. This approach yields the overall dissolution rate, including transport processes, rather than the rate of true dissolution at the mineral surface[41,42]. In particular, molecular diffusion could lead to a buildup of dissolution products next to the mineral surface, which could locally buffer seawater and raise $CaCO_3$ saturation states[43].

In volumes of undersaturated seawater, we virtually dissolve a set of foraminifera and pteropod e-specimens obtained from X-ray tomography scans (see the "Methods" section) and present micrometer-scale resolution visualizations of $CaCO_3$ saturation states and dissolution rates (Fig. 1). In each simulation, after only one minute, water inside the dissolving shells is at or near equilibrium with respect to the dissolving $CaCO_3$ phase (Fig. 1a–d). At this point, dissolution essentially only occurs on the external faces of the shells (Fig. 1e–h). The distributions of calcite dissolution rates across the foraminifera shell surfaces appear bimodal (Supplementary Fig. 2): internal faces display dissolution rates approaching zero, while external faces dissolve with rates ranging between 1.5 and $4 \times 10^{-7}$ mol m$^{-2}$ s$^{-1}$. Aragonite dissolution patterns are similar. External faces of the pteropod shell dissolve at rates between 4 and 5.5 mol m$^{-2}$ s$^{-1}$ while internal faces do not, as they are in contact with seawater at or close to equilibrium with respect to aragonite. This very wide range of values is in line with the range of calcite and aragonite dissolution rates measured in the laboratory, in seawater with a similar bulk chemical composition (Supplementary Fig. 3). Thus, part of the reason why the variability in measured dissolution rates is so large across experiments could be because solute transport, the rate-limiting step in overall dissolution, is specific to each sample and experimental design. This also shows that for a single shell with microstructures and heterogeneities, while the exposed outer surface area is dissolving, a large fraction of the total shell surface area may not be dissolving at all. When

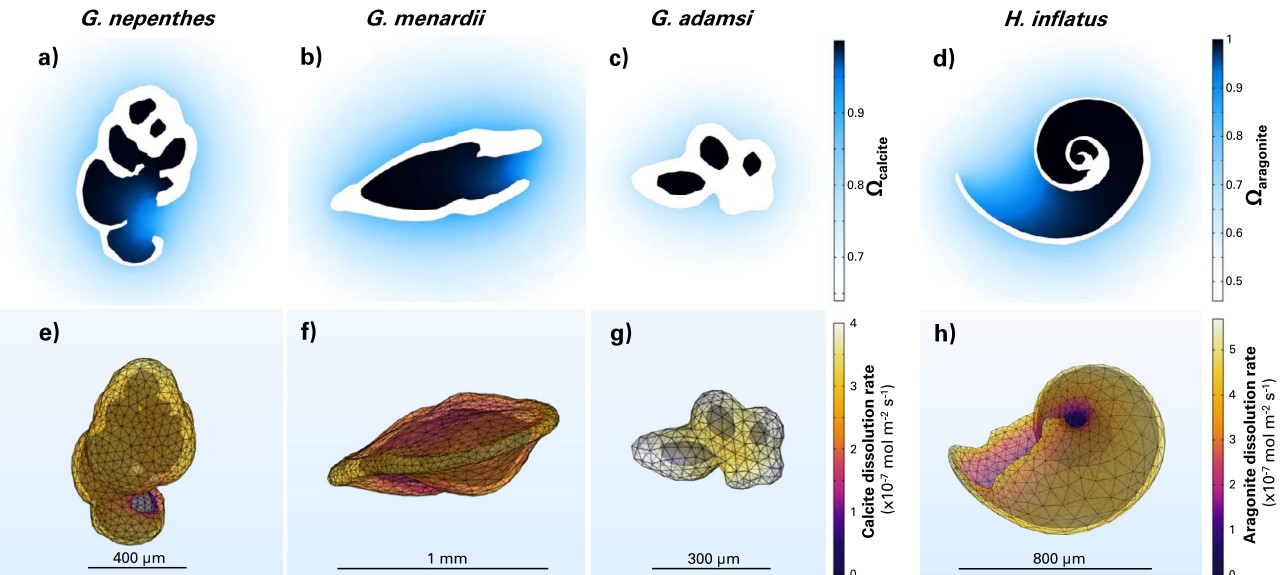

**Fig. 1 Dissolution of natural marine CaCO₃ grains after a minute in suspension in water.** The top row shows the water saturation state of calcite (**a**–**c**) and aragonite (**d**) while on the bottom row the corresponding calcite (**e**–**g**) and aragonite (**h**) dissolution rates are displayed.

expressing the overall dissolution rate of a CaCO₃ grain, its mass, rather than its surface area, may be a better property of normalization.

The specific surface areas of the foraminifera and pteropod e-specimens used in these simulations, i.e., their surface area per mass unit, are one to two orders of magnitude smaller than specific surface areas measured from the same species using the Kr-BET method[31,38,44,45]. The spatial resolution of our e-specimens is possibly not high enough to capture submicroscale features such as surface roughness and shell microporosity. Since our model underestimates mineral surfaces available for reactions, it also likely underestimates how quickly equilibrium can be reached within dissolving shells and minimizes local transport limitations. In our simulation conditions and in the absence of water advection, only the external faces of CaCO₃ shells should dissolve, as the inner parts will be at or close to equilibrium. In the following, we therefore replace calcite foraminifera shells by calcite spheres (Supplementary Table 2) of similar diameter for simplicity.

**Pteropod shell dissolution at the seafloor.** Upon death, pteropods settle rapidly (a few hundred meters per day[46]) and therefore spend only a few hours or days in the water column. Once at the deep seafloor, where typical deep-sea sediments accumulate much slower (a few centimeters per thousand years[47]), pteropods should spend a much greater time at or just below the sediment–water interface (a few decades or centuries, unaccounting for bioturbation and dissolution) than in the water column and, thus, play a role in early diagenesis of surrounding particles. We simulate the dissolution of an empty pteropod shell placed on a calcite sediment bed overlain by seawater undersaturated with respect to both calcite and aragonite ($\Omega_{calcite}$ ~ 0.64, $\Omega_{aragonite}$ ~ 0.46, Supplementary Table 1). Each calcite particle in this sediment is a sphere with a 150 µm-radius, surrogate for a typical foraminifera. The sediment bed is overlain by a 1.5-mm-thick diffusive boundary layer (Supplementary Fig. 1), within which solutes are transport via molecular diffusion. These conditions are typical of deep-sea benthic environments[30]. The dissolution simulations were run for 5 min, until a steady state was reached.

In a pure-calcite sediment bed, porewaters reach equilibrium with respect to calcite a few hundred µm below the sediment–water interface (Fig. 2) and most of the $\Omega_{calcite}$ gradient is within the diffusive boundary layer rather than the sediment (Fig. 2). This is in agreement with results from previous modeling[48,49] and laboratory[42] works on calcite-rich sediments depleted of organic-matter and aragonite. In this classical setting, the chemical gradients should be laterally homogeneous, and lead to an efflux of dissolution products from the sediment toward the bottom waters. The top layer of calcite grains should dissolve until another layer settles in, and the fraction of the calcite grains that escaped dissolution is buried, eventually, and preserved in the sediment record.

Using the same framework but replacing four calcite spheres at the sediment–water interface by an aragonitic pteropod (Supplementary Fig. 1), chemical gradients appear very different (Fig. 2). In this simulation, integrating vertically over the first layer of grains, only ~6% of the horizontal surface area (3.15 mm × 3.15 mm ≈ 10 mm², see the "Methods" section) is aragonite, the rest is calcite (32%) and water (62%). In the depth transect across the dissolving pteropod, water $\Omega_{calcite}$ increases from ~0.64 at the top of the diffusive boundary layer to ~1.3 at about 200 µm below the sediment–water interface, before decreasing again deeper in the porewaters and converging toward equilibrium (Fig. 2). Horizontally averaging $\Omega_{calcite}$ over the entire sediment mesh, we find that porewaters are saturated with respect to calcite all the way up to the sediment–water interface due to the presence of the dissolving pteropod shell (Supplementary Fig. 4). In this setting, dissolution products diffuse from the pteropod shell upward to the bottom waters, but also downward and sideways, and a halo of calcite supersaturation develops in the porewaters beneath the dissolving aragonite (Fig. 2). This causes the calcite grains surrounding the pteropod to be partially in contact with supersaturated water, thermodynamically preventing their dissolution, despite the bottom waters overlaying this sediment being strongly undersaturated with respect to calcite. Over the entire resolved domain, calcite grains sitting at the sediment–water interface only dissolve on their upper half (Fig. 3) with dissolution rates always lower than those from single-foraminifera simulations (Figs. 1 and 3).

The predicted seawater calcite supersaturation that surrounds dissolving aragonite particles at the seafloor could account for some of the calcite recrystallization occasionally observed on the

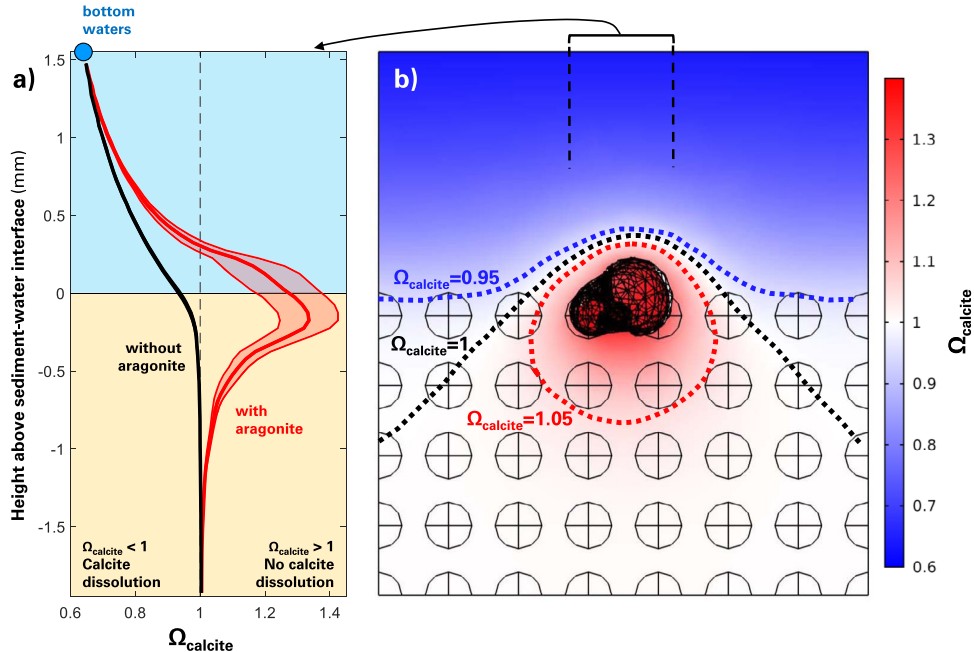

**Fig. 2 Effects of the dissolution of a pteropod shell on the saturation state with respect to calcite across the sediment–water interface. a** Depth profile of the saturation state with respect to calcite. The blue circle represents the bottom-water value. The black depth profile stands for a case without aragonite, the red depth profile represents the situation with aragonite shown on the (**b**) panel. Each depth profile is computed as the mean amongst all data points within the central 850 µm × 850 µm column, which corresponds to the size of the pteropod shell, plus and minus one standard deviation. The extent of the colored envelope surrounding the mean profiles stands for the standard deviation. **b** Depth transect of water saturation state with respect to calcite, with contours for three selected saturation state values.

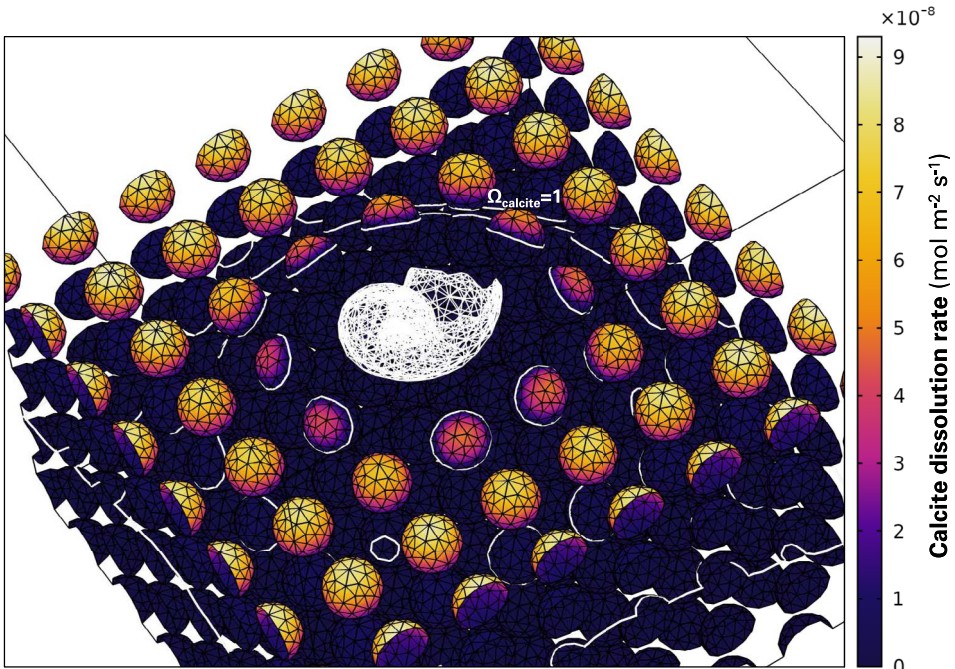

**Fig. 3 Dissolution of calcite grains in a sediment bed capped with a dissolving aragonite pteropod.** The pteropod is shown in a white mesh. Color gradients indicate surface calcite dissolution rates. The white lining represents a saturation state with respect to calcite of unity, i.e., a transition from undersaturation to supersaturation with respect to calcite that is caused by the dissolving aragonite pteropod.

surface of preserved foraminifera[50,51]. This mechanism could thus require reconsideration of the contribution of authigenic $CaCO_3$ formation to the total $CaCO_3$ burial rate in sediments, thought to be ~10%[52] and mainly due to deeper diagenetic processes such as bacterial sulfate reduction[52,53]. In addition, the loss of aragonite during taphonomy has been recognized by others as skewing community structures[54,55]. Our results suggest that taphonomic aragonite loss could lead to taphonomic calcite gain. Aragonite-based calcite galvanization is consistent with observations from the Australian continental shelf[56], in which

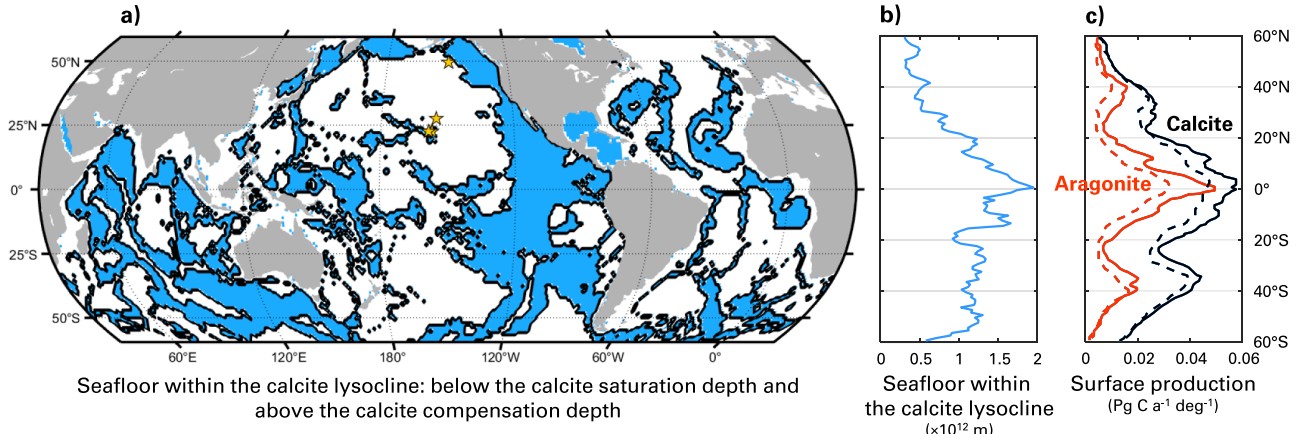

Seafloor within the calcite lysocline: below the calcite saturation depth and above the calcite compensation depth

**Fig. 4 Likely main loci of aragonite-based calcite galvanization. a** Map showing in blue the surface area of seafloor that is located below the current calcite saturation depth and the preindustrial calcite compensation depth, i.e., the calcite lysocline; both fields are from ref. [30]. The four yellow stars in the North Pacific correspond to the three sites where ref. [31] observed aragonite in the water column below the aragonite saturation depth and the site where ref. [33] measured pteropod genetic material in bottom waters. **b** Zonal integral of the seafloor surface area within the calcite lysocline. **c** Zonal integrals of the surface productions of aragonite (in red) and calcite (in black) in the uppermost 100 m of the surface ocean in ESM2M (solid) and ESM2G (dashed), averaged over 100 yr after a 1000 yr spinup; taken from ref. [17].

presumably organic-matter degradation-driven aragonite dissolution is associated with calcite preservation and reprecipitation.

**Implications for CaCO₃ cycling**. Aragonite-based calcite galvanization should occur mainly in areas of the seafloor where calcite is abundant and is dissolving in marine sediments. This mainly corresponds to seafloor areas located below the calcite saturation depth and above the calcite compensation depth, within the calcite lysocline[57], which represents about 40% of the seafloor (Fig. 4).

Aragonite can only play a meaningful role in benthic calcite dynamics via its galvanizing action if the aragonite flux to and the residence time at the seafloor are high enough. Unfortunately, little is known about the sources and sinks of aragonite in the ocean. To our knowledge, the only in situ measurements reporting the presence of aragonite[31] or pteropod genetic material[33] in deep waters are from the North Pacific (Fig. 4). In the modern surface ocean, published estimates of the contribution of pteropod aragonite to global CaCO₃ production span a very wide range from ~10% to ~90%[9,10,16]; this does not account for aragonite produced by heteropods and benthic organisms. GFSL's ESM2 models[17], which include aragonite, predict aragonite surface production to be the largest near the Equator (Fig. 4), where the surface area of seafloor found within the calcite lysocline is also the largest. In addition, it is thought that aragonite-producing pteropods are abundant in high-latitude systems[11]. At the seafloor, the world-averaged CaCO₃ deposition rate is estimated to range between 0.08 and 0.14 mol m$^{-2}$ a$^{-1}$ [7], and we cannot exclude the possibility that a substantial fraction of that amount is deposited in the form of aragonite.

It would take about twice as long for a pteropod to dissolve at the seafloor than when falling through the water column (Supplementary Fig. 5). The preferential preservation of pteropods at the sediment–water interface is due to the strong transport limitation dictated by molecular diffusion within sediment porewaters and the diffusive boundary layer above the bed, and to the presence of dissolving calcite spheres beneath it. It takes ~200 days to fully dissolve an ~800 μg aragonite sphere (Supplementary Table 2) at the seafloor (Supplementary Fig. 5). A ~60 μg pteropod shell (Supplementary Table 2) should, by extrapolation, fully dissolve in ~15 days. Given that one pteropod shell is able to maintain ~10 mm² of calcite seafloor (super)

saturated with respect to calcite all the way up to the sediment–water interface (Supplementary Fig. 4), at least one new pteropod shell needs to be delivered every 15 days to every 10 mm² of seafloor to sustain galvanization by aragonite. Using 60 μg as a typical pteropod shell weight, this translates into an aragonite deposition rate of ~0.14 mol m$^{-2}$ a$^{-1}$, on the higher end of the world-averaged total CaCO₃ deposition rate to the seafloor. Thus, it is likely that, locally, calcite particles are preferentially preserved due to aragonite dissolution at the seafloor.

Diagenetic processes excluded from the present abiotic model could affect the results presented here in various ways. On the one hand, microbial degradation of organic matter that releases CO₂ and drives additional CaCO₃ dissolution[26,58,59] should reduce the residence time of pteropod shells and other aragonite grains at the seafloor, thus hindering their galvanizing action. On the other hand, biological mixing caused by bioturbating organisms should transport aragonite grains from the sediment–water interface to depth, favouring aragonite preservation and disseminating aragonite "buffering pills" within the sediment. More broadly, our results highlight the need for future model-, field- and laboratory-based studies about marine CaCO₃ dynamics to consider the presence of several carbonate minerals simultaneously, with different compositions and structures, as they not only passively coexist but chemically interact with each other.

Finally, the proposed aragonite galvanization could act as another negative feedback mechanism regulating the Earth climate. Marine aragonite producers are particularly vulnerable to ocean acidification[60] which impedes the formation of aragonite shells[61,62] and promotes its dissolution[63]. In an ocean acidification episode such as that of the Anthropocene, a reduced aragonite transfer to the deep ocean may weaken the proposed galvanizing action. Calcite dissolution at the sediment–water interface will thus have to cover a greater share of the CO₂ neutralization which would lead to stronger saturation horizon and compensation depth variations.

## Methods

**Model**. All simulations were performed in COMSOL Multiphysics®, using the PARDISO solver and a Backward-Euler time stepping method. Eight dissolved species (H⁺, OH⁻, H₂CO₃*, HCO₃⁻, CO₃²⁻, Ca²⁺, Na⁺, Cl⁻) and two solid species (calcite, aragonite) were included. For each dissolved species, initial concentrations were determined using the PHREEQC software[64] at 25 °C, for a water density of 1023.6 kg m$^{-3}$ and a total alkalinity of 1950 μmol kg$^{-1}$, so that the

resulting saturation state of water with respect to calcite ($\Omega_{calcite}$) was about 0.64, a value typical of the deep sea (Supplementary Table 1; ref. [65]).

Three basic carbonate-system reactions were assumed to be instantaneous, implemented as follows:

$$H_2CO_3^* \leftrightarrow H^+ + HCO_3^- \quad (R1), \quad K_1 = \frac{a_{H^+}\, a_{HCO_3^-}}{a_{H_2CO_3^*}} \quad (1)$$

$$HCO_3^- \leftrightarrow H^+ + CO_3^{2-} \quad (R2), \quad K_2 = \frac{a_{H^+}\, a_{CO_3^{2-}}}{a_{HCO_3^-}} \quad (2)$$

$$H_2O \leftrightarrow H^+ + OH^- \quad (R3), \quad K_w = a_{H^+}\, a_{OH^-} \quad (3)$$

where, $K_1$, $K_2$, and $K_w$ are equilibrium constants of the reactions at 25 °C, set to $K_1 = 4.5 \times 10^{-7}$, $K_2 = 4.78 \times 10^{-11}$ (ref. [66]), and $K_w = 1 \times 10^{-14}$. $a_i$ is the activity of the $i$th species, computed as the product of its concentration ($c_i$) and total activity coefficient ($\gamma_i$), the latter being obtained from PHREEQC (Supplementary Table 1).

Calcite and aragonite dissolution were implemented as per:

$$Calcite \rightarrow Ca^{2+} + CO_3^{2-} \quad (R4), \quad K_{spcalcite} = a_{Ca^{2+}}\, a_{CO_3^{2-}} \quad (4)$$

$$Aragonite \rightarrow Ca^{2+} + CO_3^{2-} \quad (R5), \quad K_{sparagonite} = a_{Ca^{2+}}\, a_{CO_3^{2-}} \quad (5)$$

where $K_{spcalcite}$ is the solubility constant of calcite, taken here as $10^{-8.480}$, and $K_{sparagonite}$ is the solubility constant of aragonite, set at $10^{-8.336}$ (ref. [66]). CaCO$_3$ reactions are not instantaneous, but instead occur with associated rates that depend on solution chemistry and on the nature of the mineral. For calcite dissolution, we use kinetics from ref. [67], who identified three main pathways for the dissolution of calcite:

$$k_1 : CaCO_3(s) + H^+(aq) = Ca^{2+}(aq) + HCO_3^-(aq) \quad (6)$$

$$k_2 : CaCO_3 + H_2CO_3^* = Ca^{2+}(aq) + 2HCO_3^-(aq) \quad (7)$$

$$k_3 : CaCO_3(s) = Ca^{2+}(aq) + CO_3^{2-}(aq) \quad (8)$$

The rates of these reversible reactions were combined into a single dissolution rate law:

$$R_{calcite}[\text{mol m}^{-2}\text{s}^{-1}] = (k_1\, a_{H^+} + k_2 a_{CO_2(aq)} + k_3\, a_{H_2O})(1 - 10^{0.67\log_{10}(\Omega_{calcite})}) \quad (9)$$

where $k_1 = 8.64 \times 10^{-5}$, $k_2 = 4.78 \times 10^{-7}$ and $k_3 = 2.34 \times 10^{-9}$ are the reaction rate constants at 25 °C[68] and $a_{H_2O}$ is set to unity. $\Omega_{calcite}$ is the saturation state of water with respect to calcite defined as the ratio of the ion activity product (product of $a_{Ca^{2+}}$ and $a_{CO_3^{2-}}$) and the solubility constant of calcite. The same expression was used to compute the dissolution rate of aragonite, but $\Omega_{calcite}$ was replaced by $\Omega_{aragonite}$. This is a substantial simplification, as in reality both aragonite and calcite have specific dissolution kinetics. Nevertheless, recent laboratory experiments in seawater showed that, when normalized to the mineral surface area and for similar seawater saturation states with respect to the dissolving phase, aragonite dissolves at rates similar to calcite, if not slower[31,69]. This contrasts with earlier experiments[38] reporting very fast aragonite dissolution rates, but based on synthetic rather than biogenic aragonite and overestimated estimates of aragonite solubility[6,70]. Given that the dissolution rates derived from our model encompass measured dissolution rates at similar bulk seawater saturation states (Supplementary Fig. 3), the simplified kinetic treatment applied here should be acceptable as a first approximation, and should be replaced by a more accurate mechanistic kinetic scheme developed for dissolution in seawater-type solutions when available.

To simulate the reactive-transport of each dissolved species in water, advection-diffusion-reaction equation is implemented:

$$\frac{\partial c_i}{\partial t} + \nabla \cdot (-D_i \cdot \nabla c_i) + u \cdot \nabla c_i = R_i \quad (10)$$

where $t$ is the time (s), $\nabla$ is the three-dimensional space derivative operator nabla, $D_i$ is the diffusion coefficient (m$^2$ s$^{-1}$) of the $i$th species, $u$ is the prescribed water laminar velocity (m s$^{-1}$) and $R_i$ is the reaction input (mol m$^{-3}$ s$^{-1}$) of the $i$th species.

**Grains**. A set of CaCO$_3$ particles was used in this study, some with shapes derived from natural grains, some more conceptual with simplified geometries; their properties are summarized in Supplementary Table 2. Planktonic foraminifera shell scans of *Globoturborotalita nepenthes*, a Miocene Pacific species[71], *Globorotalia menardii*, a Pleistocene Carribean specimen[72] and *Globigerinella adamsi*, from the modern Pacific[73], were all obtained from the Tohoku University Museum *e*-foram database (http://webdb2.museum.tohoku.ac.jp/e-foram/). The *Heliconoides inflatus* pteropod shell, provided by Dr. Rosie Oakes, was obtained from a CT scan of a specimen caught at 150 m-depth in a sediment-trap located in the Cariaco Basin, in the Venezuelan shelf[74]. To make computations easier, the *H. inflatus* and *G. adamsi* e-specimens resolutions were reduced from a number of faces (triangles) of 1,969,997 and 1,107,096 for the original files, respectively, to 19,860 and 2224 for the final geometry files imported in COMSOL. Volumes and surface areas for each grains were computed in MATLAB from the output.stl geometry files, using the

Geometry and Mesh toolbox. Weights were computed by multiplying the volume of each grain by its density. The specific surface area (SSA) was computed as the surface area to mass ratio (see Supplementary Table 2).

**Simulations**. For the purposes of the present study, nine simulations were performed in total, each with different settings (Supplementary Table 3). All simulation experiments and their results are made available on Zenodo (https://doi.org/10.5281/zenodo.5741613).

First, four dissolution simulations were performed on natural CaCO$_3$ grains kept static (i.e. in suspension) in water (Supplementary Fig. 1). "Periodic" boundary conditions were applied on the external walls of water volumes, which forces concentrations on each wall to be equal to those on the opposite wall. The goal was to observe how fast each grain dissolves and what are the effects on solution chemistry within and outside the shell.

In order to quantify the effect of aragonite dissolution on porewater chemistry, two subsequent simulations were performed on CaCO$_3$ grains packed in a sediment bed (Supplementary Fig. 1), one with calcite grains only and another with calcite grains and one aragonite pteropod shell. For simplicity, each calcite particle in this sediment was a sphere with a 150 μm-radius, surrogate for a typical foraminifera[75], evenly spaced so that the total porosity of this sediment is ~0.84, typical of a deep-sea sediment[76]. This array of calcite spheres was then placed within a $3.15 \times 3.15 \times 3.5$ mm$^3$ (length × width × height, Supplementary Table 3, Supplementary Fig. 1) water cube, in which the bottom 1.95 mm were filled up with calcite spheres, the top 1.55 mm consisted of free water, and the sediment–water interface was located between the two. A "no flux" boundary condition was implemented at the bottom, "periodic" boundaric conditions on the sides, and on the top panel solute concentrations were fixed to their initial values.

Finally, three simulations were performed with a moving mesh, to estimate the grain size decrease due to dissolution for aragonite grains in three different environmental settings (in suspension, sinking, and in sediments, Supplementary Fig. 1). To minimize computational costs, the dissolving aragonite grain in these simulations was a sphere. The simulation with the sinking grain was performed by applying a prescribed laminar water flow velocity on the z-axis of $u = 100$ m day$^{-1}$, a typical sinking speed for a pteropod[46]. Bottom and top boundary conditions were set to "no flux", and boundary conditions on the sides were "periodic". In each simulation, a displacement rate, normal to the aragonite grain surface, was assigned to the aragonite reactive walls, computed as:

$$w_n = R_{aragonite}\, MV \quad (11)$$

where $w_n$ is the displacement rate defined at aragonite surface and MV is molar volume of aragonite set to $3.42 \times 10^{-5}$ m$^3$ mol$^{-1}$.

## Data availability

Original foraminifera CT scans used in this study are available from the Tohoku University Museum e-foram database (http://webdb2.museum.tohoku.ac.jp/e-foram/). The pteropod CT scan is available on request to Dr. Rosie Oakes.

## Code availability

All simulation experiments and their results are made available on Zenodo (https://doi.org/10.5281/zenodo.5741613).

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

## Acknowledgements

We thank Dr. Rosie Oakes at the Met Office and Dr. Osamu Sasaki at the Tohoku University Museum for providing CT scans of pteropods and foraminifera. We thank Julien Sulpis for his assistance in processing three-dimensional geometry files. O.S. and J.J.M. were supported by the Dutch Ministry of Education via the Netherlands Earth System Science Centre (NESSC). The research work of P.A., and M. Wolthers is part of the Industrial Partnership Program i32 Computational Sciences for Energy Research that is carried out under an agreement between Shell and the Netherlands Organization for Scientific Research (NWO). M. Wolthers has received funding from the European Research Council (ERC) under the European Union's Horizon 2020 research and innovation program (grant agreement No. [819588]). G.M. is a Research Associate with the Belgian Fund for Scientific Research F.R.S.-FNRS. Financial support for the work of G.M. was provided by the Belgian Fund for Scientific Research—F.R.S.-FNRS (project SERENATA, grant CDR J.0123.19).

## Author contributions
O.S., P.A., M.Wolthers, G.M. and J.J.M. designed the research. O.S. and P.A. performed the simulations. O.S. and M.Walker processed the geometry files. O.S. wrote the manuscript with contributions from all authors.

## Competing interests
The authors declare no competing interests.
