## [Peer Review File · Nature Communications]

REVIEWER COMMENTS

Reviewer #1 (Remarks to the Author):

This is a novel and creative study assessing the potential for sinking aragonite to preserve calcite at certain depths at the seafloor. The modeling of saturation/dissolution kinetics on particles informed by X-ray tomography of shells is impressive and demonstrates the plausibility of the process. These results are important for how calcium carbonate is modeling in global climate models and I support publication after some revisions. While the study is quite micro-focused on the dissolution process, my more substantive comments below relate to assessing some of the macro-implications of the work, and I encourage the authors to provide some more context of how this new process fits into global ocean as much as far as that is possible at this point.

Where geographically should aragonite-based calcite galvanization be occurring? I assume it is seafloor depths that are deeper than the calcite saturation horizon but shallower than the carbonate compensation depth. How much of the ocean is that? A map showing the geographic locations with this feature may be very informative as to the degree of global importance, especially in regard to the paper's prediction about future changes in carbon burial.

It would also be good to be transparent about the geographic locations of the evidence for aragonite sinking below the aragonite saturation horizon. From the Dong et al., and Boeuf et al. references cited, it seems this phenomenon could be something specific to the North Pacific? Or is the claim it should be more prevalent than that?

Distinctions between deep ocean and shelf seas. "oceanic" used on line 32. "open ocean" used on line 38. Practically, it would be helpful to provide your definition of open ocean versus coastal seas (is it deeper than 1 km depth? 200 m depth?). This is important because one of the studies cited (Milliman, 1993) established the idea that in the whole ocean, only about half of the carbonate produced is in the deep ocean—the shelf seas being a huge producer of carbonate, including reefs and large carbonate platforms. It would be good to be transparent about this too.

Line 132-133. The terms "very slow accumulation rates" and "spending a much greater time" are all relative. Compared to the red clays of the North Pacific, a few cm per thousand years could be called quite a high sedimentation rate. It would be better to rephrase this sentence using the actual assumed sedimentation rates and burial times with justification for those assumptions, and avoiding the relative terms.

Line 156. The term "sediment-water (horizontal) interface surface area" is still a little mysterious to me. If minerals are being integrated in a mass balance, there must be some depth scale over which the integration is done, no? What is the thickness assumed for the sediment-water interface?

Line 213-214 "delivered every 15 days, every 10 mm²" needs to be clarified or rephrased. Does this mean for every 10 mm² patch, 1 pteropod must be delivered every 15 days?

Reviewer #3 (Remarks to the Author):

The manuscript "Aragonite is calcite's best friend at the seafloor" by Sulpis et al. describes the mitigating role that easily dissolvable forms of calcium carbonate play in the preservation of less dissolvable calcite through the use of a novel combination of 3D imaging and modeling of sediment dissolution. While I have several minor criticisms detailed below, I find the study highly creative and compelling, well worthy of publication in Nature Communications.

title - "best friend" seems a bit unprofessional and not directly appropriate as the relationship the author's

describe is not so much symbiotic peer to peer "friendship" but sacrificial, much like sacrificial zinc used to protect steel. I don't have a good alternative suggestion though.

13 – "10-90%" is a nine-fold range... I think the calcium carbonate cycle is known a little better than that from the synthesis assessment of Iglesias-Rodriguez et al., 2002, Progress Made in Study of Ocean's Calcium Carbonate Budget, *Eos*, Vol. 83, No. 34,

<https://agupubs.onlinelibrary.wiley.com/doi/epdf/10.1029/2002EO000267> and Schiebel, R. (2002). Planktic foraminiferal sedimentation and the marine calcite budget. *Global Biogeochemical Cycles*, 16(4), 3-1.

33 – This assertion ignores that the IPSL/PISCES and GFDL models have included aragonite (the first of which is cited later in the manuscript). A statement like "While a few global modeling studies have included aragonite, ..." is warranted:

Gangstø, R., Gehlen, M., Schneider, B., Bopp, L., Aumont, O., & Joos, F. (2008). Modeling the marine aragonite cycle: changes under rising carbon dioxide and its role in shallow water CaCO₃ dissolution. *Biogeosciences*, 5(4), 1057-1072.

Dunne, J. P., John, J. G., Shevliakova, E., Stouffer, R. J., Krasting, J. P., Malyshev, S. L., ... & Zadeh, N. (2013). GFDL's ESM2 global coupled climate-carbon earth system models. Part II: carbon system formulation and baseline simulation characteristics. *Journal of Climate*, 26(7), 2247-2267.

Stock, C. A., Dunne, J. P., Fan, S., Ginoux, P., John, J., Krasting, J. P., ... & Zadeh, N. (2020). Ocean Biogeochemistry in GFDL's Earth System Model 4.1 and its Response to Increasing Atmospheric CO₂. *Journal of Advances in Modeling Earth Systems*, 12(10), e2019MS002043.

The potential role of high magnesium calcite cycling should also be mentioned here and elsewhere:

Wilson, R. W., Millero, F. J., Taylor, J. R., Walsh, P. J., Christensen, V., Jennings, S., & Grosell, M. (2009). Contribution of fish to the marine inorganic carbon cycle. *Science*, 323(5912), 359-362.

Woosley, R. J., Millero, F. J., & Grosell, M. (2012). The solubility of fish-produced high magnesium calcite in seawater. *Journal of Geophysical Research: Oceans*, 117(C4).

45 – "should dissolve" meaning "at equilibrium". This contrasts with calcite preservation which should also dissolve at equilibrium but is often kinetically preserved (Archer, D. (1991). Modeling the calcite lysocline. *Journal of Geophysical Research: Oceans*, 96(C9), 17037-17050.)

63 – "since it is the most soluble mineral present" is true only if high magnesium calcite is ignored or otherwise deemed unimportant.

64 – "picture" should be "conceptual model"

110 – Given the above description of the importance of distinguishing "external" high versus "internal" negligible rates, it is not clear to me why the authors would then conclude "When expressing the overall dissolution rate of a CaCO₃ grain, its mass, rather than its surface area, may be a better property of normalization." More explanation is necessary.

213 – "15 days," should be "15 days to"

214 – "in order" is extraneous

Reviewer #1 (Remarks to the Author):

This is a novel and creative study assessing the potential for sinking aragonite to preserve calcite at certain depths at the seafloor. The modeling of saturation/dissolution kinetics on particles informed by X-ray tomography of shells is impressive and demonstrates the plausibility of the process. These results are important for how calcium carbonate is modeling in global climate models and I support publication after some revisions. While the study is quite micro-focused on the dissolution process, my more substantive comments below relate to assessing some of the macro-implications of the work, and I encourage the authors to provide some more context of how this new process fits into global ocean as much as far as that is possible at this point.

We thank the reviewer for their time and constructive feedback. We have added more context to describe the implications of our study, and hope this will address the concern of the reviewer:

- We have modified Fig. 4 and replaced it with a map and a zonal plot showing where the aragonite-based calcite galvanization should be most important, and a zonal plot showing aragonite versus calcite production predictions from two Earth System models. We have also added a paragraph describing this new figure and the areas of the seafloor where implications of our studies are most important
 - [L209-L212]: “Aragonite-based calcite galvanization should occur mainly in areas of the seafloor where calcite is abundant and is dissolving in marine sediments. This mainly corresponds to seafloor areas located below the calcite saturation depth and above the calcite compensation depth, within the calcite lysocline⁵⁷, which represents about 40% of the seafloor (Fig. 4).”
 - The new Fig. 4 is shown below

Figure 4. Likely main loci of aragonite-based calcite galvanization. a) Map showing in blue the surface area of seafloor that is located below the current calcite saturation depth and the preindustrial calcite compensation depth, i.e., the calcite lysocline; both fields are from ref. (30). The four yellow stars in the North Pacific correspond to the three sites where ref. (31) observed aragonite in the water column below the aragonite saturation depth and the site where ref. (33) measured pteropod genetic material in bottom waters. b) Zonal integral of the seafloor surface area within the calcite lysocline. c) Zonal integrals of the aragonite (in red) and calcite (in

35 *black) pumps within the in the upper 100 m for ESM2M (solid) and ESM2G (dashed) averaged
over 100 yr after over 1000 yr of spinup; taken from ref. (17).*

- We have added some text and references regarding earlier work about the taphonomic loss of aragonite, which supports aragonite-based calcite galvanization
 - 40 ○ [L196-L200] *“In addition, the loss of aragonite during taphonomy has been recognized by others as skewing community structures^{54,55}. Our results suggest that taphonomic aragonite loss could lead to taphonomic calcite gain. Aragonite-based calcite galvanization is consistent with observations from the Australian continental shelf⁵⁶, in which presumably organic-matter degradation-driven*
45 *aragonite dissolution is associated with calcite preservation and reprecipitation.”*
- We have added a paragraph at the end of the main text describing the implications of our new results for predictions of future ocean acidification
 - 50 ○ *“Finally, the proposed aragonite galvanization could act as another negative feedback mechanism regulating the Earth climate. Marine aragonite producers are particularly vulnerable to ocean acidification⁶⁰ which impedes the formation aragonite shells^{61,62} and promotes its dissolution⁶³. In an ocean acidification episode such as that of the Anthropocene, a reduced aragonite transfer to the*
55 *deep ocean may weaken the proposed galvanizing action. Calcite dissolution at the sediment-water interface will thus have to cover a greater share of the CO2 neutralization which would lead to stronger saturation horizon and compensation depth variations.*

60 Where geographically should aragonite-based calcite galvanization be occurring? I assume it is seafloor depths that are deeper than the calcite saturation horizon but shallower than the carbonate compensation depth. How much of the ocean is that? A map showing the geographic locations with this feature may be very informative as to the degree of global importance, especially in regard to the paper’s prediction about future
65 changes in carbon burial.

We agree with the reviewer that aragonite-based calcite galvanization should be occurring mainly in areas of the seafloor that are below the calcite saturation depth and above the calcite compensation depth, which corresponds to seafloor areas that are within the calcite lysocline. We
70 have taken the current calcite saturation depth and pre-industrial calcite compensation depth distributions from ref. (30) and used them to plot these areas on a map. According to these estimates, about 40% of the seafloor is within the calcite lysocline. We have added this map and new information in the revised manuscript, see the first bullet point in the response to the reviewer #1 first comment.

75 It would also be good to be transparent about the geographic locations of the evidence for aragonite sinking below the aragonite saturation horizon. From the Dong et al., and Boeuf

et al. references cited, it seems this phenomenon could be something specific to the North Pacific? Or is the claim it should be more prevalent than that?

80

We have added the four sites from the Dong et al. and Boeuf et al. studies in which aragonite or pteropod genetic material was found below saturation depth as yellow stars on the new Fig. 4; all are in North Pacific. We have also added the following sentence:

85

- [L215-217]: *“To our knowledge, the only in situ measurements reporting the presence of aragonite³¹ or pteropod genetic material³³ in deep waters are from the North Pacific (Fig. 4).”*

90

Distinctions between deep ocean and shelf seas. “oceanic” used on line 32. “open ocean” used on line 38. Practically, it would be helpful to provide your definition of open ocean versus coastal seas (is it deeper than 1 km depth? 200 m depth?). This is important because one of the studies cited (Milliman, 1993) established the idea that in the whole ocean, only about half of the carbonate produced is in the deep ocean—the shelf seas being a huge producer of carbonate, including reefs and large carbonate platforms. It would be good to be transparent about this too.

95

To clarify, we have added the following:

- [L29-L30]: In the open ocean, *“that we define here as all oceanic areas beyond continental shelves”*, [...]

100

Line 132-133. The terms “very slow accumulation rates” and “spending a much greater time” are all relative. Compared to the red clays of the North Pacific, a few cm per thousand years could be called quite a high sedimentation rate. It would be better to rephrase this sentence using the actual assumed sedimentation rates and burial times with justification for those assumptions, and avoiding the relative terms.

105

The “much greater time” spent by pteropods at the seafloor was relative to the water column. We have precised that, and also added numbers and references for the typical accumulation rates. This portion of the text now reads as:

110

- [L138-143]: *“Upon death, pteropods settle rapidly (a few hundred meters per day⁴⁶) and therefore spend only a few hours or days in the water column. Once at the deep seafloor, where typical deep-sea sediments accumulate much slower (a few centimeters per thousand years⁴⁷), pteropods should spend a much greater time at or just below the sediment-water interface (a few decades or centuries, unaccounting for bioturbation and dissolution) than in the water column and, thus, play a role in early diagenesis of surrounding particles.”*

115

120

Line 156. The term “sediment-water (horizontal) interface surface area” is still a little mysterious to me. If minerals are being integrated in a mass balance, there must be some depth scale over which the integration is done, no? What is the thickness assumed for the sediment-water interface?

This sentence was confusing, we should not have referred to this surface as the “sediment-water (horizontal) interface surface area”, but instead simply as the “horizontal surface area” when integrated over the depth scale that corresponds to the first layer of grains. We will rephrase to:

- [L174-L176]: “*In this simulation, integrating vertically over the first layer of grains, only ~6% of the horizontal surface area ($3.15\text{ mm} \times 3.15\text{ mm} \approx 10\text{ mm}^2$, see Materials and Methods) is aragonite, the rest is calcite (32%) and water (62%).*”

Line 213-214 “delivered every 15 days, every 10 mm^2 ” needs to be clarified or rephrased. Does this mean for every 10 mm^2 patch, 1 pteropod must be delivered every 15 days?

Yes, we have rephrased according to reviewer #3’s suggestion and hope it clarifies this sentence.

Reviewer #3 (Remarks to the Author):

The manuscript “Aragonite is calcite’s best friend at the seafloor” by Sulpis et al. describes the mitigating role that easily dissolvable forms of calcium carbonate play in the preservation of less dissolvable calcite through the use of a novel combination of 3D imaging and modeling of sediment dissolution. While I have several minor criticisms detailed below, I find the study highly creative and compelling, well worthy of publication in Nature Communications.

We thank the reviewer for their time and constructive feedback.

Title - "best friend" seems a bit unprofessional and not directly appropriate as the relationship the author's describe is not so much symbiotic peer to peer "friendship" but sacrificial, much like sacrificial zinc used to protect steel. I don't have a good alternative suggestion though.

The “friendship” between calcite and aragonite is in this case indeed one-sided, but we feel that it is best to avoid jargon and being too technical in the title, hence lacking a better alternative we will stick to this title.

13 – “10-90%” is a nine-fold range... I think the calcium carbonate cycle is known a little better than that from the synthesis assessment of Iglesias-Rodriguez et al., 2002, Progress Made in Study of Ocean's Calcium Carbonate Budget, Eos, Vol. 83, No. 34, <https://agupubs.onlinelibrary.wiley.com/doi/epdf/10.1029/2002EO000267> and Schiebel, R. (2002). Planktic foraminiferal sedimentation and the marine calcite budget. Global Biogeochemical Cycles, 16(4), 3-1.

We agree that this range is very wide and the true contribution of aragonite to open-ocean CaCO₃ export is probably halfway between these extremes, but this reflects the range of published estimates so far. In the abstract we have replaced this portion of the text by “whose contribution to global pelagic calcification could be at par with that of calcite”. In the second paragraph of the section “Implications for CaCO₃ cycling”, we cite the studies from which this range was taken. In this revised manuscript, we have included a new figure that includes a zonal plot of aragonite versus calcite production in the surface ocean as predicted by the GFDL ESM2M and ESM2G. In the new Fig. 4, according to predictions from ESM2M and ESM2G, the aragonite fraction within the CaCO₃ surface production is at par with that of calcite at the Equator, and globally, between 30 and 60% of all open-ocean CaCO₃ production is aragonite.

33 – This assertion ignores that the IPSL/PISCES and GFDL models have included aragonite (the first of which is cited later in the manuscript). A statement like “While a few global modeling studies have included aragonite, ...” is warranted:

Gangstør, R., Gehlen, M., Schneider, B., Bopp, L., Aumont, O., & Joos, F. (2008). Modeling the marine aragonite cycle: changes under rising carbon dioxide and its role in shallow water CaCO₃ dissolution. *Biogeosciences*, 5(4), 1057-1072.

Dunne, J. P., John, J. G., Shevliakova, E., Stouffer, R. J., Krasting, J. P., Malyshev, S. L., ... & Zadeh, N. (2013). GFDL’s ESM2 global coupled climate–carbon earth system models. Part II: carbon system formulation and baseline simulation characteristics. *Journal of Climate*, 26(7), 2247-2267.

Stock, C. A., Dunne, J. P., Fan, S., Ginoux, P., John, J., Krasting, J. P., ... & Zadeh, N. (2020). Ocean Biogeochemistry in GFDL’s Earth System Model 4.1 and its Response to Increasing Atmospheric CO₂. *Journal of Advances in Modeling Earth Systems*, 12(10), e2019MS002043.

We have included the suggested sentence addition and references.

The potential role of high magnesium calcite cycling should also be mentioned here and elsewhere:

Wilson, R. W., Millero, F. J., Taylor, J. R., Walsh, P. J., Christensen, V., Jennings, S., & Grosell, M. (2009). Contribution of fish to the marine inorganic carbon cycle. *Science*, 323(5912), 359-362.

Woosley, R. J., Millero, F. J., & Grosell, M. (2012). The solubility of fish-produced high magnesium calcite in seawater. *Journal of Geophysical Research: Oceans*, 117(C4).

We have rephrased this portion of the text to include the suggested references, the text now reads as:

- [L35-L37]: “In addition, magnesium (Mg) calcites, which can be twice more soluble as aragonite¹², are also thought to be important in the open ocean, secreted by fish^{12,13} or imported from shallow shelves and banks^{14,15}.”

We have also added another sentence with these references elsewhere:

- [L70-L72]: “In practice, the possible presence of Mg calcites^{12,13} at the seafloor could complicate this model further, and the dissolution of Mg calcites protect aragonite from dissolution.”

215 45 – “should dissolve” meaning “at equilibrium”. This contrasts with calcite preservation which should also dissolve at equilibrium but is often kinetically preserved (Archer, D. (1991). Modeling the calcite lysocline. Journal of Geophysical Research: Oceans, 96(C9), 17037-17050.)

220 Below their saturation depths, both aragonite and calcite should dissolve, but if grains sink or accumulate fast, or if external processes are slowing dissolution down, then calcite and aragonite can indeed be preserved.

225 63 – “since it is the most soluble mineral present” is true only if high magnesium calcite is ignored or otherwise deemed unimportant.

225 This is true. We have added this information, see the response to the comment above.

225 64 – “picture” should be “conceptual model”

230 We agree, and have edited the text accordingly.

230 110 – Given the above description of the importance of distinguishing “external” high versus “internal” negligible rates, it is not clear to me why the authors would then conclude “When expressing the overall dissolution rate of a CaCO₃ grain, its mass, rather than its surface area, may be a better property of normalization.” More explanation is necessary.

235 We have added an extra sentence to clarify:

- [L115-L117]: *“This also shows that for a single shell with microstructures and heterogeneities, while the exposed outer surface area is dissolving, a large fraction of the total shell surface area may not be dissolving at all.”*

240 213 – “15 days,” should be “15 days to”

245 We agree and have updated the text.

245 214 – “in order” is extraneous

250 We have deleted “in order”.

REVIEWERS' COMMENTS

Reviewer #1 (Remarks to the Author):

As I mentioned in my first review, I am supportive of this manuscript and believe it is a valuable addition to the understanding of calcium carbonate cycling with important implications in the global ocean. The authors had perfect responses and associated revisions to my comments and those of the other reviewer. I especially like Fig. 4 showing the geographical implications very clearly. I support publication with only some minor changes for grammar, listed below.

Line 44, add colon after "due to" or add numbers to the listed factors (the factors are fairly long so this helps the reader keep them straight).

Line 61, add "a" before "surrogate"

Line 167, Fig. 2 caption, "enveloped" should be "envelope"

Line 173, "much different" is not grammatical. Replace with "very different" or "considerably different" or something along those lines.

Line 193, replace "to reconsider" with "reconsideration of"

Line 213, replace "calcite benthic dynamics" with "benthic calcite dynamics"

Line 221, replace "comprised" with "found"

Reviewer #1 (Remarks to the Author):

As I mentioned in my first review, I am supportive of this manuscript and believe it is a valuable addition to the understanding of calcium carbonate cycling with important implications in the global ocean. The authors had perfect responses and associated revisions to my comments and those of the other reviewer. I especially like Fig. 4 showing the geographical implications very clearly. I support publication with only some minor changes for grammar, listed below.

We thank Reviewer #1 for their positive feedback.

Line 44, add colon after "due to" or add numbers to the listed factors (the factors are fairly long so this helps the reader keep them straight).

We have added numbers to the listed factors.

Line 61, add "a" before "surrogate"

We have added that.

Line 167, Fig. 2 caption, "enveloped" should be "envelope"

We have corrected that.

Line 173, "much different" is not grammatical. Replace with "very different" or "considerably different" or something along those lines.

We have replaced "much" by "very".

Line 193, replace "to reconsider" with "reconsideration of"

We have replaced that.

Line 213, replace "calcite benthic dynamics" with "benthic calcite dynamics"

We have replaced that.

Line 221, replace "comprised" with "found"

We have replaced that.